# DRIMET: Deep Registration for 3D Incompressible Motion Estimation in Tagged-MRI with Application to the Tongue

**Zhangxing Bian** [1]                                   ZBIAN4@JHU.EDU
**Fangxu Xing** [2]                                    FXING1@MGH.HARVARD.EDU
**Jinglun Yu** [1]                                      JYU146@JHU.EDU
**Muhan Shao** [1]                                     MUHAN@JHU.EDU
**Yihao Liu** [1]                                      YLIU236@JHU.EDU
**Aaron Carass** [1]                               AARON_CARASS@JHU.EDU
**Jiachen Zhuo** [3]                                    JZHUO@UMM.EDU
**Jonghye Woo** [2]                                  JWOO@MGH.HARVARD.EDU
**Jerry L. Prince** [1]                                  PRINCE@JHU.EDU

[1] *Electrical and Computer Engineering, Johns Hopkins University, USA*

[2] *Radiology, Massachusetts General Hospital and Harvard Medical School, USA*

[3] *Diagnostic Radiology and Nuclear Medicine, University of Maryland School of Medicine, USA*

**Editors:** Accepted for publication at MIDL 2023

## Abstract

Tagged magnetic resonance imaging (MRI) has been used for decades to observe and quantify the detailed motion of deforming tissue. However, this technique faces several challenges such as tag fading, large motion, long computation times, and difficulties in obtaining diffeomorphic incompressible flow fields. To address these issues, this paper presents a novel unsupervised phase-based 3D motion estimation technique for tagged MRI. We introduce two key innovations. First, we apply a sinusoidal transformation to the harmonic phase input, which enables end-to-end training and avoids the need for phase interpolation. Second, we propose a Jacobian determinant-based learning objective to encourage incompressible flow fields for deforming biological tissues. Our method efficiently estimates 3D motion fields that are accurate, dense, and approximately diffeomorphic and incompressible. The efficacy of the method is assessed using human tongue motion during speech, and includes both healthy controls and patients that have undergone glossectomy. We show that the method outperforms existing approaches, and also exhibits improvements in speed, robustness to tag fading, and large tongue motion. The code is available: https://github.com/jasonbian97/DRIMET-tagged-MRI.

**Keywords:** Tagged MRI, incompressible, deep learning, registration, motion estimation, Jacobian determinant

## 1. Introduction

Estimating the motion of soft tissues undergoing deformation is a major topic in medical imaging research. Magnetic resonance (MR) tagging (Axel and Dougherty, 1989a,b) has a wide range of applications in diagnosing and characterizing coronary artery disease (McVeigh et al., 1998; Edvardsen et al., 2006), cardiac imaging (Kolipaka et al., 2005; Ibrahim, 2011), speech and swallowing research (Parthasarathy et al., 2007; Xing et al., 2019; Gomez et al., 2020), and brain motion in traumatic injuries (Knutsen et al., 2014). MR tagging temporarily

magnetizes tissue with a spatially modulated periodic pattern, creating transient tags in the image sequence that move with the tissue, thus capturing motion information.

The harmonic phase (HARP) method (Osman et al., 1999, 2000; Osman and Prince, 2000) is a widely-used technique for processing tagged-MR. HARP computes phase images from sinusoidally tagged MR images using bandpass filters in the Fourier domain. Based on the fact that the harmonic phases of material points do not change with motion, HARP tracking uses harmonic phase images as proxies in the image registration framework to avoid the violation of brightness consistency caused by tag fading. However, interpolating phase values during registration can be challenging, as it requires local phase unwrapping (Xing et al., 2017). Also, the unwrapping operation is not differentiable, making it difficult to leverage in an end-to-end learning framework. Global phase unwrapping (Spoorthi et al., 2018; Wang et al., 2022) is one possible solution to this problem, but it has been found to be error-prone when MR imaging artifacts exist (Jenkinson, 2003). In this paper, we propose a sinusoidal transformation for harmonic phases that eliminates the need for phase interpolation or phase unwrapping, while still being resistant to imaging artifacts and tag fading. Given this transformed input, we designed an unsupervised multi-channel image registration framework for a set of 3D tagged images with different tag orientations.

In this study, we focus on estimating tongue motion. According to the American Cancer Society, approximately 48,000 people in the US are diagnosed with oral or oropharyngeal cancer annually, and 33% of these cases affect the tongue (American Cancer Society, 2016). Understanding the motion differences between healthy subjects and those who have undergone a glossectomy can help inform surgical decisions and assist in speech and swallowing remediation. Compared to extensively-studied cardiac motion, tongue motion has four unique properties that make motion estimation challenging: 1) the tongue moves quickly relative to the temporal resolution of the scan during speech; 2) the motion is aperiodic and highly variable among subjects; 3) the tongue is highly deformable during speech due to its interdigitated muscle architecture (Abd-El-Malek, 1955); and 4) the air gap present in the oral cavity can significantly affect the quality of imaging, causing severe artifacts. To address these issues, we developed an unsupervised deep learning model that utilizes phase information of tagged MR images (MRIs) to estimate the motion field. Our model has shown superior performance on healthy subject data and has also demonstrated good generalization to patients who have undergone a glossectomy.

Incompressibility is another crucial factor to consider when analyzing biological movements. For example, research has shown that the volume change of the myocardium during a cardiac cycle is less than 4% (Yin et al., 1996), and the volume change of the tongue during speech and swallowing is even smaller (Gilbert et al., 2007). Therefore, it is necessary to assume that muscle motion is incompressible in order to accurately represent the physical tissue properties (Kier and Smith, 1985).

**Contributions**    1) We propose a sinusoidal transformation for harmonic phases, which is resistant to noise, artifacts, and tag fading, and can be easily incorporated into the end-to-end training of modern deep learning-based registration frameworks. 2) We propose a determinant-based objective for learning 3D incompressible flow that better represents the motion of human biological tissue. To the best of our knowledge, this is the first work that learns to estimate incompressible motion fields within a deep learning-based image registration framework. 3) With the aforementioned two features, we propose a novel

unsupervised deep learning-based method called DRIMET, to directly estimate 3D dense incompressible motion fields on tagged MRI. Our approach is robust to tag fading, large motion, and can generalize well to pathological data.

## 2. Background & Related Work

**Tagging MRI-based 3D motion estimation**    Tracking 3D motion is generally necessary when estimating the motion of biological structures. MR tagging is one way of imaging motion (Chitiboi and Axel, 2017). In the past, traditional 2D MR tagging motion estimation methods have been extended to 3D (Ryf et al., 2002; Abd-Elmoniem et al., 2007; Spottiswoode et al., 2008). However, these methods require the acquisition of a large number of closely spaced image slices, making them impractical for routine clinical use due to the large amount of time required. Other approaches estimate 3D motion directly from sparse imaging geometries, such as using finite element or finite difference methods (O'Dell et al., 1995), tag line tracking based on 2D images (Denney and Prince, 1995; Ye et al., 2021), or spline interpolation (Denney and Prince, 1995; Huang et al., 1999). These methods typically require a 3D tissue segmentation, which can be time-consuming and may require human intervention or automated segmentation algorithms. PVIRA (Xing et al., 2017) proposed to interpolate tag images to form a finer grid for later tracking and also uses HARP magnitude images to eliminate the need for a segmentation model. However, PVIRA uses phase images for registration, requiring local phase unwrapping during interpolation, which can be error-prone and is non-differentiable. In contrast, we propose a novel sinusoidal transformation for 3D harmonic phase images, thus avoiding the need for phase unwrapping and allowing for differentiable interpolation techniques such as trilinear interpolation. By incorporating this transformation, we can achieve end-to-end training and improved accuracy.

**Deep learning-based image registration**    Iterative optimization approaches (Avants et al., 2011; Vercauteren et al., 2009; Mansi et al., 2011) have been successful in achieving good accuracy in intensity-based deformable image registration. However, these methods optimize for each new image pair and can be slow. Alternatively, deep learning-based registration and optical flow approaches can directly take a pair of images (or multi-frames) as input and output the corresponding estimated deformations, resulting in faster inference speeds and comparable accuracy to iterative methods (Balakrishnan et al., 2019; Mok and Chung, 2020; Liu et al., 2022; Chen et al., 2022; Jia et al., 2022; Hering et al., 2022; Bian et al., 2022). Such frameworks learn the function $g_\theta(F, M) = \phi$ which gives the transformation $\phi$ that is used to align the fixed $F$ and moving image $M$. The function $g$ is parametrized by learnable parameters $\theta$. The parameters are learned by optimizing the generalized objective:

$$\hat{\theta} = \arg\min_{\theta} \mathcal{L}_{\text{sim}}(F, M \circ g_\theta(F, M)) + \lambda \mathcal{L}_{\text{smooth}}(g_\theta(F, M)). \qquad (1)$$

The first term, $\mathcal{L}_{\text{sim}}$, encourages image similarity between the fixed and warped moving image. The second term, $\mathcal{L}_{\text{smooth}}$, imposes a smoothness constraint on the transformation. The parameter $\lambda$ determines the trade-off between these two terms. These approaches typically try to match two scalar images—one fixed and one moving. In contrast, our method is designed to simultaneously match three pairs of tagged images with different tag orientations, which is critical for solving the aperture problem. Additionally, these

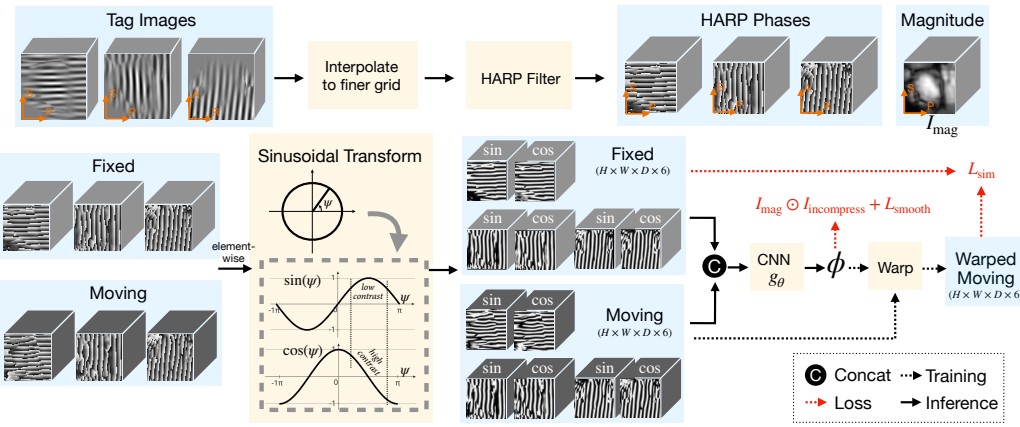

Figure 1: Top: HARP processing pipeline. Bottom: HARP phases of fixed and moving images are taken as input. They undergo a sinousoidal transformation and are sent into a UNet-like multi-channel registration network.

approaches do not preserve the incompressibility of the motion field, whereas our approach estimates high-quality incompressible flow fields while being much faster.

**Incompressiblity**    Incompressible flow fields, also known as divergence-free vector fields or volume-preserving transformations, have been a longstanding research topic in fluid dynamics (Majda and Bertozzi, 2002) and image registration (Song and Leahy, 1991; Gorce et al., 1997). In this paper, we focus on their application to biological image registration. Previously, there have been two main types of approaches: iterative approaches (Mansi et al., 2009, 2011; Xing et al., 2017; Yu et al., 2023) and determinant-based approaches (Rappoport et al., 1995; Rohlfing et al., 2003; Haber and Modersitzki, 2004). Iterative approaches incorporate incompressibility into diffeomorphic demons (Vercauteren et al., 2009) when updating the stationary velocity fields iteratively, making them computationally expensive. Determinant-based approaches focus on constraining the deformation field with a penalty on the Jacobian determinant of the deformation deviating from unity. This assumes that tissue can be deformed locally, but the volume (local and total) remains approximately constant. In the past, the Jacobian determinant has been used as an incompressibility regularization term to constrain coordinate transformation during B-spline-based nonrigid image registration (Rohlfing et al., 2003). However, this constraint is non-linear and requires ad-hoc numerical schemes that are computationally demanding (Haber and Modersitzki, 2004; Mansi et al., 2011). Recent work (Mok and Chung, 2020) enforces orientation consistency of the deformation field using a determinant constraint for diffeomorphism while leaving incompressilibility unsolved. In contrast, we introduce a novel Jacobian determinant-based learning objective into the unsupervised deep learning-based registration framework and show its effectiveness in preserving volume and achieving a diffeomorphism. Furthermore, our learned model takes less than a second to process a single pair of frames, compared to tens of minutes required by previous works.

## 3. Method

**HARP processing** The harmonic phase (HARP) algorithm is a well-established method for processing and analyzing tagged MRIs (Osman et al., 2000). Specifically, HARP filtering involves extracting the first spectral peak in the Fourier domain of a tagged image slice to obtain a complex-valued image, where the phase part (HARP phase) contains motion information and the magnitude part (HARP magnitude) contains anatomical information. Since tagged images are typically acquired with lower through-plane resolution, they are interpolated onto a finer isotropic grid before HARP processing (Xing et al., 2017). Figure 1 shows the 3D tagged image processing using interpolation and HARP. Our method applies HARP filtering to the three interpolated tag volumes $I_{\mathrm{Sh}}$, $I_{\mathrm{Sv}}$, and $I_{\mathrm{Av}}$. For example, for the vertically-tagged axial volume $I_{\mathrm{Av}}(\boldsymbol{x})$, the complex image $J_{\mathrm{Av}}(\boldsymbol{x})$ after HARP filtering is computed as $J_{\mathrm{Av}}(\boldsymbol{x}) = D_{\mathrm{Av}}(\boldsymbol{x})e^{j\Psi_{\mathrm{Av}}(\boldsymbol{x})}$, where $D_{\mathrm{Av}}$ is the HARP magnitude volume and $\Psi_{\mathrm{Av}}$ is the HARP phase volume. The same notation applies for the horizontally- and vertically-tagged sagital volumes, yielding $D_{\mathrm{Sh}}$, $\Psi_{\mathrm{Sh}}$, $D_{\mathrm{Sv}}$, and $\Psi_{\mathrm{Sv}}$. We average over the three maginitude images to obatain $I_{\mathrm{Mag}}$.

**Sinousoidal transform** Phase-based registration is recognized as more robust than intensity-based registration for dealing with tag fading, geometric distortions between frames, and noise (Fleet and Jepson, 1993; Hemmendorff et al., 2002). Interpolating phase values requires local phase unwrapping when the phase difference between two points exceeds $\pi$ (Xing et al., 2017). However, phase unwrapping is non-differentiable, which is a problem for deep learning-based registration methods, whose training typically rely on backpropagation.

To address this, we propose a simple sinusoidal transformation for phase images. Specifically, given a phase image $\Psi$, we apply element-wise sin and cos operations,

$$I^{\sin}(\boldsymbol{x}) = \sin(\Psi(\boldsymbol{x})) \quad \text{and} \quad I^{\cos}(\boldsymbol{x}) = \cos(\Psi(\boldsymbol{x})), \tag{2}$$

to obtain two corresponding images $(I^{\sin}, I^{\cos})$. Thus a phase value is uniquely associated with a $(\sin, \cos)$ pair and vice versa, i.e., one-to-one mapping. This transformation allows for smooth interpolation while still retaining the robustness of phase-based registration to tag fading, distortions between frames, and noise. Additionally, the flat (low-contrast) regions and steep (high-contrast) region in sin and cos are compensated for by each other, as demonstrated in Figure 1 (in "sinusoidal transform" module). An alternative way to view this sinusoidal transformation is by writing the complex image as $J(\boldsymbol{x}) = D(\boldsymbol{x})(\cos(\Psi(\boldsymbol{x})) + j\sin(\Psi(\boldsymbol{x})))$, where the subscript is omitted. Thus, $(I^{\cos}, I^{\sin})$ can be seen as the real and imaginary parts of the complex image $J(\boldsymbol{x})/D(\boldsymbol{x})$.

**Multi-channel registration** Harmonic phase is a material property that can be used to solve the aperture problem in optical flow by tracking the three harmonic phase values that come from three linearly independent tag directions. Instead of tracking phase values directly, however, because of the one-to-one nature of the sinusoidal transformation, we can track the patterns in the sinusoidally transformed image pairs. Note that here we match *multiple* fixed and moving sinusoidal images at the same time. To do this, we used the following mean squared error (MSE) as our similarity loss during training:

$$\mathcal{L}_{\mathrm{sim}} = \sum_{k \in \{\mathrm{Av,Sh,Sv}\}} \sum_{l \in \{\sin,\cos\}} \mathrm{MSE}(F_k^l, M_k^l \circ \boldsymbol{\phi}), \tag{3}$$

where

$$\phi = g_\theta \left( \texttt{concat} \left\{ F_k^l, M_k^l \mid k \in \{\mathrm{Av}, \mathrm{Sh}, \mathrm{Sv}\}, l \in \{\sin, \cos\} \right\} \right) . \tag{4}$$

**Incompressible constraint**    Volume preservation, also known as incompressibility, is an important feature for image registration in moving biological tissues. Accordingly, we introduce the following Jacobian determinant-based learning objective on the transformation field to encourage incompressiblity:

$$\mathcal{L}_{\mathrm{incompress}} = \sum_{\boldsymbol{x}} I_{\mathrm{Mag}}(\boldsymbol{x}) \left| \log \max \left( |J_\phi(\boldsymbol{x})|, \epsilon \right) \right| - \sum_{\boldsymbol{x}} \min \left( |J_\phi(\boldsymbol{x})|, 0 \right) , \tag{5}$$

where $\epsilon$ is a small positive number and $I_{\mathrm{Mag}}$ is the HARP magnitude image with a range of [0,1]. $I_{\mathrm{Mag}}$ serves as a soft mask for tissues such as the tongue. The first term penalizes the deviation of the Jacobian determinant $|J_\phi(\boldsymbol{x})|$ from unity and is spatially weighted by $I_{\mathrm{Mag}}$. With the logarithm, the proposed loss *symmetrically* penalize expansion and contraction. We introduced a second term in Equation (5) both to prevent the solution where all determinants are negative and to encourage a diffeomorphism by directly penalizing negative determinants. The proposed loss encourages $\phi$ to be incompressible in tissue regions and diffeormophic everywhere including in air gaps. While the L1 and L2 penalties $||J_\phi(\boldsymbol{x})| - 1|_{\{1,2\}}$ are also viable, they were found to be less effective than Equation (5).

**Overall training objective**    We encourage the spatial smoothness of the displacement $\boldsymbol{u}$, with the smoothness loss $\mathcal{L}_{\mathrm{smooth}} = \sum_{\boldsymbol{x}} \|\nabla \boldsymbol{u}(\boldsymbol{x})\|^2$. Thus *the overall loss* for training is $\mathcal{L}_{\mathrm{total}} = \mathcal{L}_{\mathrm{sim}} + \lambda \mathcal{L}_{\mathrm{smooth}} + \beta \mathcal{L}_{\mathrm{incompress}}$, where $\lambda$ and $\beta$ are hyper-parameters.

## 4. Experiments

**Materials**    The present study includes a dataset of 25 unique (subject-phrase) pairs, consisting of 8 healthy controls (HCs) and 2 post-partial glossectomy patients with tongue flaps. To capture tongue movement during speech, the participants were asked to say one or more phrases—"a thing", "a souk", or "ashell"—while tagged MRIs were acquired. The recorded phrases had a duration of 1 second, with 26 time frames captured during this period. The in-plane resolution of the MR images is $1.875 \times 1.875$ mm, and the slice thickness is 6 mm. The tagged MR images were collected using the CSPAMM pulse sequence in both sagittal (both vertical and horizontal tags) and axial (vertical tags only) orientations. The HC data was split (subject-phrase)-wise into training, validation, and test datasets in a ratio of 6:2:2; the post-partial glossectomy patient data was reserved for testing.

**Network architecture**    As our goal is to explore the value of the proposed sinousoidal transformation and incompressible objective in a generic setting rather than combining advanced techniques to achieve state-of-the-art, we used the well-established 3D U-Net (Ronneberger et al., 2015) architecture with a convolutional kernel of size $(5 \times 5 \times 5)$ (Jia et al., 2022). To accommodate our multi-channel setting, we set the input channel of the first convolutional kernel to 12 to accept 6-channels from each of the fixed and moving sinusoidal images—sin and cos for each of the three acquired images. A scaling and squaring layer (Dalca et al., 2018) is used as the final layer to encourage diffeomorphism. To test the scalibility of our model, we trained a larger model, termed "Ours-L", by doubling the number of intermediate feature channels.

Table 1: Quantitative measurement of performance on registration accuracy (w/ RMSE), incompressiblity (w/ Det_AUC), diffeomorphims (w/ NegDet), and speed (w/ Time). The p-values of Wilcoxon signed-rank tests between "Ours" and others are reported.

| | Registration Acc: RMSE ↓ | | | Incompressibility: Det_AUC ↑ | | | NegDet (%) ↓ | Time ↓ |
|---|---|---|---|---|---|---|---|---|
| | mean ± std | median | $p$ | mean ± std | median | $p$ | mean ± std | (s/pair) |
| PVIRA | 0.153 ± 0.053 | 0.160 | <0.001 | 0.936 ± 0.031 | 0.935 | <0.001 | 0.000 ± 0.005 | 49 |
| Ours w/o inc. | 0.122 ± 0.041 | 0.126 | <0.001 | 0.862 ± 0.077 | 0.870 | <0.001 | 0.019 ± 0.039 | <0.1 |
| Ours | 0.132 ± 0.045 | 0.137 | – | **0.950 ± 0.038** | **0.956** | – | **0.000 ± 0.000** | <0.1 |
| Ours-L | **0.122 ± 0.038** | **0.126** | <0.001 | 0.950 ± 0.039 | 0.956 | <0.001 | 0.000 ± 0.001 | <0.1 |

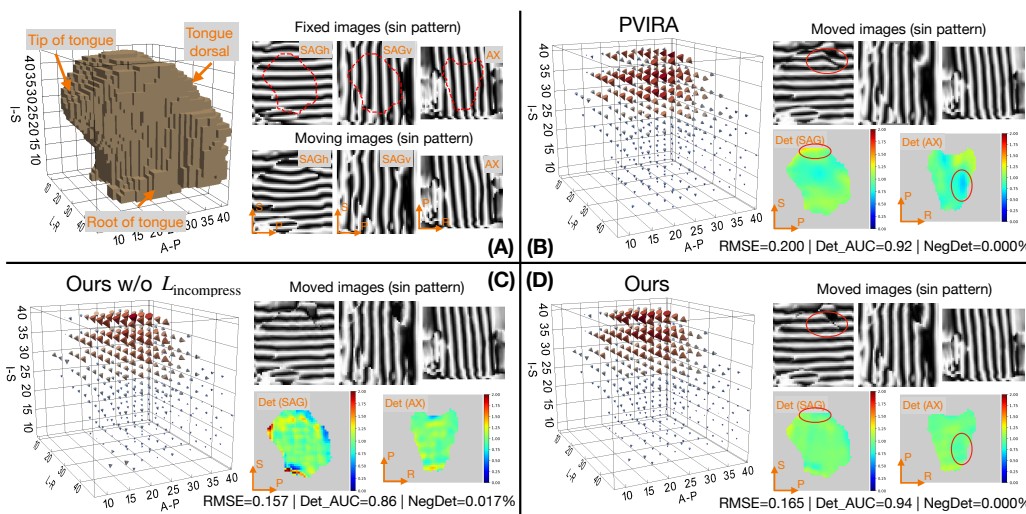

Figure 2: **(A)** An example of a fixed and moving frame pair is shown, with only the sin pattern displayed (the cos pattern is omitted). The contour of the tongue region is annotated in the fixed image with a red dotted line. **(B-D)** Results of three different methods are shown, including 3D motion field, the warped moving images, and a sagittal and an axial slice of the 3D Jacobian determinant map. Major differences are highlighted with red circles.

**Training details**  During training, fixed and moving image frames are randomly selected from a speech sequence, with the maximum time gap between them being 8 frames. We augment each pair of images with random center-crops during training. No overfitting is observed. The best loss weights (hyper-parameter $\lambda$, $\beta$) are determined by grid search for each model to ensure fair comparision. See more details about the materials, imaging settings, and training in Appendix A.

## 5. Results

**Registration accuracy**  It is often difficult to obtain the true dense motion field for evaluating the accuracy of registration algorithms. However, since harmonic phase is a property of tissue that moves with the tissue, we can expect that a motion field that is

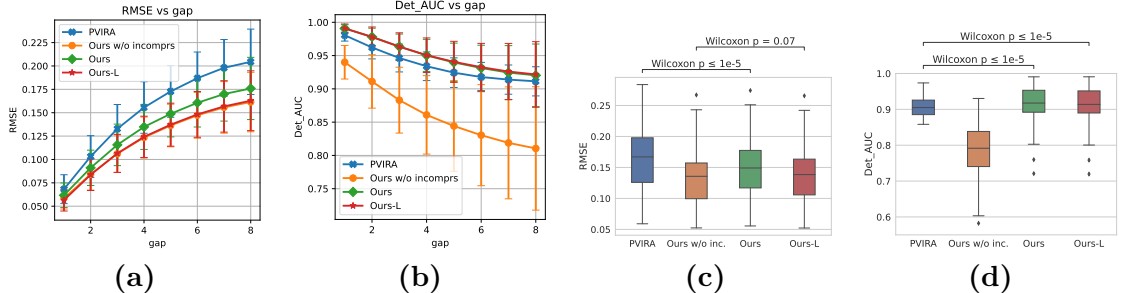

Figure 3: Shown in **(a)** and **(b)** are the performance of the various methods against large motion. While **(c)** and **(d)** show performance on the pathological cases.

closer to the true motion would result in a better voxel-wise intensity match between fixed and warped moving harmonic phase images. The sinusoidal transformation is a one-to-one mapping, so we use the root mean squared error (RMSE) between the sinusoidal-transformed fixed and warped moving images as a measure of registration accuracy.

**Incompressibility**    A perfect incompressible flow field has a $|J_\phi(\boldsymbol{x})|$ of 1 everywhere. To assess incompressibility, we compute the histogram of determinant errors (*i.e.* $||J_\phi(\boldsymbol{x})| - 1|$) and weight it by $I_{\mathrm{Mag}}$. We then calculate the area under the cumulative distribution function (CDF) curve (AUC) as a scalar metric. A higher AUC indicates a more incompressible motion field. See more explanations on Det_AUC in Appendix. B.

As shown in Table 1, The proposed method (labeled "Ours") significantly outperforms PVIRA in terms of both registration accuracy and incompressibility. Without the proposed incompressibility constraint ($\mathcal{L}_{\mathrm{incompress}}$), The "Ours w/o inc." model fails to preserve incompressibility and is non-diffeomorphic in some instances. However, it still performs well in terms of registration accuracy, indicating a general trade-off between these two factors. The same trade-off is observed with VoxelMorph being the network architecture (in Appendix C). Interestingly, our larger model ("Ours-L") achieves the best registration accuracy of all the models while also maintaining nearly the same ability to estimate incompressible flow as the best model ("Ours"). Figure 2 shows an example when the subject initiates the phrase "a thing" from a neutral position, the tongue moves back and downward.

**Degradation with large time gaps**    We also evaluated the methods on pairs of frames with different time gaps, which are assumed to be proportional to the magnitude of motion (within a short time window during speech, e.g., 8-frame window). As shown in Figure 3, our model degrades less severely as the motion becomes larger. Additionally, since the effect of tag-fading accumulates with larger time gaps, these results also demonstrate the robustness of our models to tag-fading.

**Generalizability to pathological subjects**    Our model also demonstrates better performance on patients who have undergone a glossectomy, as shown in Figure 3. We note that our models were only trained on data from HCs and were then directly evaluated on patient data without any fine-tuning. This demonstrates the generalizability of our models, despite their being trained on a different population. Due to the unsupervised nature, we see further improvement with instance-specific optimization on patient data (in Appendix D).

**Limitations**    While our approach demonstrated superior performance in large motion compared to other methods, there remains large room for improvement. Our current experiment was conducted solely on tongue data. Nevertheless, we believe it can be

generalized to other organs, as the sinusoidal patterns we used are independent of the specific underlying anatomy. An error analysis is included in Appendix E.

## 6. Conclusion

We proposed a sinusoidal transformation and determinant-based objective for unsupervised estimation of dense 3D incompressible motion fields from tagged MRI. The method is robust to tag fading and large motion, and can generalize well to pathological data. We believe that the success of our preliminary tongue motion study indicates the potential of our proposed techniques for cardiac and brain motion tracking with tagged MRI.

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

## Appendix A. Experiment details

### A.1. Material details

All participants in the study, which was approved by the University of Maryland Baltimore Institutional Review Board (protocol number: HP-00042060), gave written informed consent. The eight healthy controls, five men and three women, were between 24 and 32 years old, while the two patients, both women, were 38 and 40 years old. All subjects were native speakers of American English.

### A.2. Imaging Settings

The study used a 3T Prisma MR scanner (Siemens Healthcare) and a 64-channel head/neck coil to obtain images of the tongue's motion during speech. Participants repeated a phrase while tagged MR images were acquired with a field of view of $240 \times 240$ mm$^2$, TE = 1.47 ms, and TR = 36 ms. Each data set includes a stack of images covering the entire tongue and surrounding tissues. Multiple repetitions of the speech task were performed to collect tagged data. The speech task was timed to a metronome repeated every 2 seconds.

### A.3. Training Details

We set $\lambda = 0.01$ and $\beta = 0.4$ for the models denoted as "Ours" and "Ours-L". We set $\lambda = 0.08$ for the "Ours w/o inc." model where the $\mathcal{L}_{\text{incompress}}$ is not applied. $\epsilon = 10^{-5}$ in $\mathcal{L}_{\text{incompress}}$. For scaling and squaring, we used 7 as the number of iterations for all model variants. In all experiments, we used the Adam optimizer with a batch size of one and a fixed learning rate of $1 \times 10^{-4}$ throughout training. For all model variants, we trained using a batch size of 1 and trained until convergence at 1M steps without observing overfitting. Training time took around 30 hours using a single Quadro RTX 5000 GPU. PVIRA was tested on a

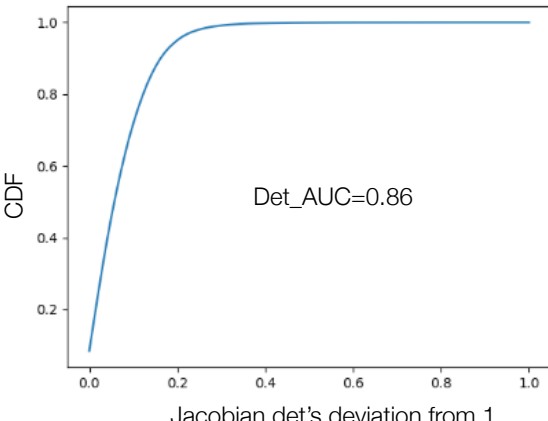

Figure 4: An example of the cumulative dirstirbution funciton (CDF) of the error of Jacobian determinants from unit. The area under the CDF curve is 0.86.

Linux machine with 20 Intel(R) Xeon(R) Silver 4208 @ 2.10GHz CPUs. The images are manually cropped to include the tongue region and then zero-padded to $64 \times 64 \times 64$. The (mean,std) of width, height, and depth of tongue region before zero-padding are: (48.3, 4.2), (56.5, 2.9), (54.2, 3.7), respectively. We note that application to other organs might necessitate retraining if the input image size or the tag spacing are different.

## Appendix B. Explaination on Det_AUC

We use Det_AUC as a metric to measure deformation field incompressibility. For a given deformation field, we compute the determinants ($Det$) at each voxel grid location and calculate the histogram (weighted by $I_{\mathrm{Mag}}$) of errors $|Det - 1|$. The CDF curve is computed from histogram and the area under the curve is taken as the metric. Intuitively, a left-leaning CDF curve with a higher AUC indicates higher incompressibility, becuase more determinants are closer to unity. An example is given in Fig. 4. We use Det_AUC metric instead of mean absolute error (MAE) because Det_AUC is less sensitive to outliers.

## Appendix C. Ablation on Network Architecture

We further explored the effectiveness of the proposed loss function with different network architectures: the popular VoxelMorph (Balakrishnan et al., 2019) and the state-of-the-art LKUnet (Jia et al., 2022). The version of VoxelMorph that was used[1] includes the implementation of scaling and squaring. The results, as shown in Table 2, demonstrate that training with our proposed loss leads to the estimation of more diffeomorphic and incompressible motion fields.

---

1. https://github.com/voxelmorph/voxelmorph/tree/dev/voxelmorph/torch

Table 2: Evaluations on different choices of network architectures and their performances with the proposed incompressible loss. The asterisk (*) indicates that the original network has undergone minimal changes to enable the acceptance of multi-channel input for our task. The reported p-values are of the Wilcoxon signed-rank test computed between "LKUnet* + Inc." and the other methods.

| | Registration Acc: RMSE ↓ | | | Incompressibility: Det_AUC ↑ | | | NegDet (%) ↓ | Time ↓ |
|---|---|---|---|---|---|---|---|---|
| Backbone | mean ± std | median | $p$ | mean ± std | median | $p$ | mean ± std | (s/pair) |
| VoxelMorph* | 0.129 ± 0.047 | 0.130 | <0.001 | 0.851 ± 0.093 | 0.862 | <0.001 | 0.015 ± 0.025 | <0.1 |
| VoxelMorph* + Inc. | 0.135 ± 0.051 | 0.137 | <0.001 | 0.946 ± 0.045 | 0.951 | <0.001 | **0.000 ± 0.000** | <0.1 |
| LKUnet* | 0.122 ± 0.041 | 0.126 | <0.001 | 0.862 ± 0.077 | 0.870 | <0.001 | 0.019 ± 0.039 | <0.1 |
| LKUnet* + Inc. | 0.132 ± 0.045 | 0.137 | – | **0.950 ± 0.038** | **0.956** | – | **0.000 ± 0.000** | <0.1 |
| LKUnet-L* + Inc. | **0.122 ± 0.038** | **0.126** | <0.001 | 0.950 ± 0.039 | 0.956 | <0.001 | 0.000 ± 0.001 | <0.1 |

## Appendix D. Instance-specific Optimization on Patient Data

Patient data is both scarce and heterogeneous. We trained our model exclusively on healthy controls and then test our model's performance on patient data. Specificly, two pathological subjects were evaluated, each with 5 different subject-phrase cases. Every subject-phrase case consists of 26 time frames.

In this section, we aim to be more practical and solution-oriented. Leveraging the unsupervised nature of our method, we conducted instance-specific optimization on each test image pair. This technique is similar to what was done in VoxelMorph (Balakrishnan et al., 2019). We conducted optimization individually for each test pair using gradient descent for 30 iterations, taking 5.6 seconds for "Ours" and 6.1 seconds for "Ours-L". Figure 5 shows instance-specific optimization achieves improved registration accuracy and comparable Det_AUC. Numerical results are available in Table 3.

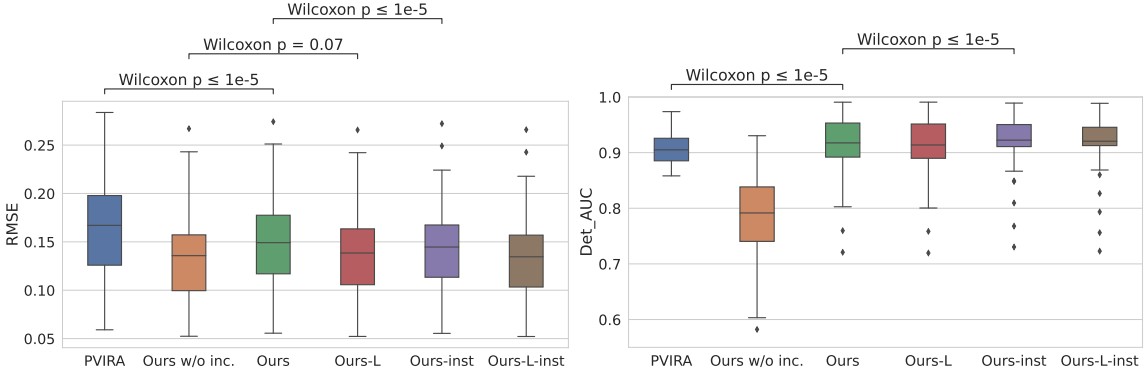

Figure 5: Performance on paitent data. The suffix "-inst" indicates the results with instance-specific optimization.

Table 3: Numerical results on paitent data. The suffix "-inst" indicates the results with instance-specific optimization.

| | Registration Acc: RMSE ↓ | | | Incompressibility: Det_AUC ↑ | | | NegDet (%) ↓ | Time ↓ |
|---|---|---|---|---|---|---|---|---|
| | mean ± std | median | $p$ | mean ± std | median | $p$ | mean ± std | (s/pair) |
| PVIRA | 0.162 ± 0.054 | 0.167 | <0.001 | 0.909 ± 0.031 | 0.904 | <0.001 | 0.000 ± 0.005 | 49 |
| Ours w/o inc. | 0.132 ± 0.045 | 0.136 | <0.001 | 0.774 ± 0.108 | 0.792 | <0.001 | 0.039 ± 0.071 | <0.1 |
| Ours | 0.146 ± 0.048 | 0.149 | − | 0.915 ± 0.056 | 0.917 | − | 0.000 ± 0.000 | <0.1 |
| Ours-L | 0.136 ± 0.047 | 0.138 | <0.001 | 0.912 ± 0.057 | 0.914 | <0.001 | 0.000 ± 0.000 | <0.1 |
| Ours-inst | 0.142 ± 0.047 | 0.145 | <0.001 | 0.918 ± 0.051 | 0.922 | <0.001 | 0.000 ± 0.000 | 5.6 ± 0.3 |
| Ours-L-inst | 0.133 ± 0.046 | 0.135 | <0.001 | 0.917 ± 0.053 | 0.920 | <0.001 | 0.000 ± 0.001 | 6.1 ± 0.4 |

## Appendix E. Error Analysis

Figure 6 depicts a representative error pattern. The determinant map illustrates non-incompressible flow in the vicinity of the tongue tip, where large motion is present. Two factors contribute to the errors. Firstly, the HARP magnitude image is an imperfect soft mask, and the constraint weighted by the soft mask may not be sufficient to generate an incompressible motion field in the proximity of the tongue boundary. Secondly, imaging artifacts introduced by the air gap cause a discontinuity in the tag line, leading to a mismatch on both sides of the tongue boundary. To mitigate these issues in future work, a possible solution is to employ a binary tongue mask to weigh the smoothness and incompressible penalties. The idea of weighting the smoothness constraint with a binary tongue mask also aligns with the "gliding" motion of the tongue in the oral cavity, where the smoothness constraint should be disregarded at the air and tissue boundary.

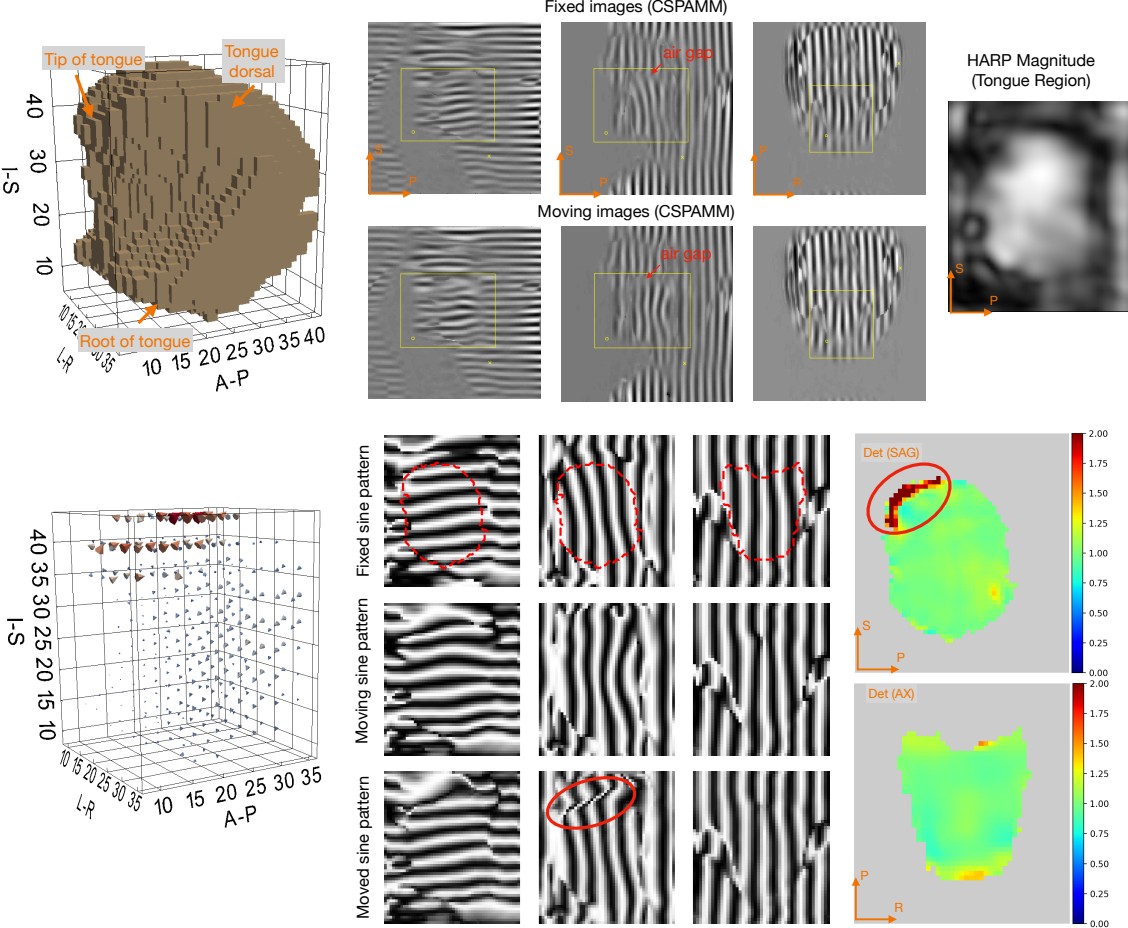

Figure 6: An example with large error. The RMSE=0.181, Det_AUC=0.913, NegDet=0.0.

