# OpenReview forum: "DRIMET: Deep Registration-based 3D Incompressible Motion Estimation in Tagged-MRI with Application to the Tongue"
_MIDL.io/2023/Conference — MIDL 2023 Oral_

### Official Review · Reviewer_LNQa · 2023-01-30

**Confidence:** 5
**Preliminary Rating:** 4
**Recommendation:** Poster

**Summary:**

This paper proposes an energy term and a deep learning model for estimating the 3d motion field from tagged mri. Using the output from the HARP algorithm, the paper proposes to use the phase volume function on Euler cos-sin form as input to a 3d-UNet, whose loss function consists of a dissimilarity term, a smoothness term, and a loosely defined incompressible term.  The method is applied to 25 images of tongue motion and is compared with a single, similar method from the literature PVIRA.

**Strengths:**

The paper is well written, modelling the incompressible is sensible, the results are better than the single method compared to from the literature. I have nothing further to add. I have nothing further to add.

**Weaknesses:**

It is difficult to assess the quality of the method without ground truth and with limited comparison with state-of-the art.  I have nothing further to add. I have nothing further to add. I have nothing further to add.


**Deanonymize Review:**

yes

**Detailed Comments:**


1. on bottom of p.3 (1) is "unable to preserve the incompressibility" - I find that a strong statement, since restricting the norm of the jacobian of the transformation has a smoothing effect.
2. (3) - the euler angles are not invertible: a=cos(b+n 2 \pi) for integer n, as stated below "and vice versa". Thus, I don't understand, how this form avoids the unwrapping problem discussed.
3. below (3) "as demonstrated in Figure 1". I don't see how this is demonstrated in the figure
4. (5) - g({...}) is a weird notation, since g does not take a set.
5. L_total above Section 4 - please highlight this final loss function
6. Materials - please discuss whether the anisotropic voxels has an effect on the result, and possibly should be accounted for in the loss function.
7. References - please add the keys to the list of references, so that keys from the text are easier to find in this list.

**Paper Type:**

methodological development

**Questions To Address In The Rebuttal:**

See above. See above. See above. See above. See above. See above. See above. See above. See above. See above. See above. See above. See above. See above. See above. See above. See above. See above. See above.

---

### Official Review · Reviewer_XbTu · 2023-02-02

**Confidence:** 3
**Preliminary Rating:** 4
**Recommendation:** Poster

**Summary:**

The authors present a deep-learning method for motion estimation in 3D based on tagged MRI data. They applied their method to tongue motion estimation with success and show that the improvements they made by using sine-transform to make images differentiable and introducing an incompressibility constraint resulted in more accurate estimation than previous methods.

**Strengths:**

The paper is well-written and the method and results are well-presented. The added features of sine transform and compressibility are very good ideas which are shown to be important for such algorithms. This method seem to have the potential to be applied on other types of similar datasets.

**Weaknesses:**

I am not an expert enough in this field of 3d estimations but I do not see any major weakness. The method is compared only against one existing method, maybe a more thorough comparison with previous similar methods would help show the strength of this approach.

**Deanonymize Review:**

no

**Detailed Comments:**

I found some typos:
- below eq (2): horizaonally- and veritcally-tagged
- above eq (4): here, must match

Also, fig.3 labels are a bit too small, they could be enlarged for better readability while keeping the figure size fit the paper format.


**Paper Type:**

both

**Questions To Address In The Rebuttal:**

Here are some questions that came to me while reading:

Why do you think that your method also works on pathological subjects while trained on healthy ones?

How much the words pronounced matter for the method to work?

Is this method directly applicable to cardiac and brain motion tracking, or some significant adjustment would have to be made?

---

### Official Review · Reviewer_ftbo · 2023-02-03

**Confidence:** 3
**Preliminary Rating:** 4
**Recommendation:** Poster

**Summary:**

This paper presents a learning-based technique for measuring the motion of deforming tissue in tagged magnetic resonance imaging. It is applied to human tongue tissue during speech in both healthy controls and patients with glossectomy.
The algorithm uses Harmonic Phase (HARP) MRI data which were generated by imprinting a fixed frequency sinusoidal spatial modulation pattern in the MRI data. This HARP pattern can be separated from the other image-related frequencies by filtering in the Fourier domain and consequently approximations of the unaltered image and the tagging pattern can be generated.  Tagging in multiple dimensions and fast repeated imaging is used to track tissue motion.
A limitation of phase information is that the absolute phase of the tagged signal can not be observed directly, only the wrapped phase which is the absolute phase modulo $2\pi$. A key component of the proposed algorithm is the projection of the wrapped HARP phase image to polar coordinates (Euler's formula) which removes phase wrap related discontinuities in the tagged phase maps. A Unet followed by a scaling and squaring operation is used to predict a warp from three orthogonal tag encodings projected to polar coordinates in a moving and a fixed image.
Muscle tissue can be approximated as incompressible and the method addresses this by penalizing deformations that locally compress or expand the tissue. The loss incorporates a penalty for non-smooth deformations and for local tissue volume changes in areas of high signal magnitude using a custom weighting of the deviation of the Jacobian determinant from unity.

**Strengths:**

The algorithm works in 3D and no ground truth motion trajectories need to be generated as the network can learn directly from the motion sequences. The use of phase images allows learning a Unet-based registration algorithm from limited data and alleviates issues due to tag fading which would require intensity models and brightness constraints.

It is fast compared to iterative registration algorithms and accurate compared to a reference algorithm "PVIRA" when evaluated and trained on data from the same population. In particular, on these data, it yields deformations with a Jacobian determinant that closer to unity than PVIRA.

**Weaknesses:**

The impact of the proposed methods is not clear. The work is specific to HARP and references to recent methods that address similar issues such as Harmonic Phase Interpolation of HARP data (DOI:10.1109/TMI.2021.3051092) are missing. What are advantages and limitations of HARP that might affect this work and how do they relate to related methods that estimate deformations such as Local Sine-Wave Modeling (DOI:10.1109/TMI.2009.2037955) and Gabor filters (DOI:10.1109/IEMBS.2006.259542)? Please give some context by contrasting to other imaging techniques that allow motion tracking such as Steady State Free Precession (DOI:10.1016/j.jcmg.2009.11.006) imaging, Displacement Encoding with Stimulated Echoes (DOI:10.1006/jmre.1998.1676) or Strain-encoded Magnetic Resonance (DOI:10.1002/jmri.21612).
Other closely related work that is not cited:
- DOI:10.1109/TMI.2011.2168825
- DOI:10.1109/TMI.2022.3154599
- DOI:10.1117/12.2610989
- DOI:10.1088/1361-6560/aad109
- DOI:10.1007/s11263-006-8984-4
- DOI:10.1118/1.3193526
- DOI:10.1109/TMI.2003.814791

The utility of the polar encoding is not clear. Does it improve performance compared to learning directly on the wrapped phase maps? For instance, PhaseNet (DOI:10.1109/LSP.2018.2879184) uses straightforward phase augmentations to learn continuity across phase wraps implicitly. This would allow halving the size of the input data.
Also, the phase mapping using polar coordinates is only a one-to-one mapping with the wrapped phase, not the absolute phase. The projection does not recover longer-range phase information and the resulting intensities are periodic. This likely causes local minima in the similarity loss for instance for translations that correspond to phase shifts beyond $2 \pi$. Techniques that use phase unwrapping to recover the absolute phase could utilise longer-range phase information. While this method alleviates the need for phase unwrapping, phase maps can be unwrapped as a preprocessing step using existing methods such as ROMEO (DOI:10.1002/mrm.28563). Furthermore, learning-based approaches could directly estimate the unwrapped phase (DOI:10.1117/1.APN.1.1.014001) which could be trained end-to-end - contrary to what is stated in the paper.

The method claims to generate incompressible deformations but incompressibility is not enforced during inference. I would encourage to authors to reword the title of the paper.

While performance on the training data appears superior to PVIRA, the validation of the method (and its parts) is lacking.
- Generalisation to other anatomies and imaging settings is unclear. If retraining is required, this should be stated as it adds to the computational expense.
- Other iterative algorithms or learning-based methods are mentioned but PVIRA is the only algorithm this method is compared against. Is PVIRA the clear gold-standard?
- Generalisation tests are limited to a small patient population. Please consider validation on other organs. For instance PVIRA used simulated phantoms, human brain and tongue data during speech, as well as public cardiac data for validation.
- The authors state that the algorithm performs favourably compared to PVIRA. Yet, assuming the metric to describe incompressibility (`Det_AUC`) is a valid metric (see Detailed Comments) and the box plots in Fig 3 d) show median and percentiles, the algorithm appears to be less robust and performs on average worse on the pathological cases compared to PVIRA on this metric.
  The RMSE of the phase maps might be affected by local minima due to the periodic nature of the projected phase maps. Please report average performance values and show images of the failure cases. You could compare the effect of the warps on the patient images to assess which method is better and comment on whether either are suitable for clinical or research use.

**Deanonymize Review:**

no

**Detailed Comments:**

- `Det_AUC`: Its definition is not complete, and the motivation for this metric is unclear and it should be defined on first use.
	- is the same histogram used for all comparisons and how is it defined?
	- consider replacing this metric by a more interpretable metric, see for instance Fig 6h) in DOI:10.1109/TMI.2017.2723021
- the claim that phase unwrapping is not differentiable is not generally true
- penalty and constraints are used interchangeably
- for a diffeomorphic warp, the Jacobian determinant needs to have constant sign (not zero), a non-negativity penalty is not sufficient
- the value of epsilon is not reported
- how large are the crops before zero padding?
- please report training requirements (time and GPU memory) and the setup used to compare runtimes of this method and PVIRA
- some of the results and discussion are stated in the background section (but not backed up with evidence): "whereas our approach estimates an incompressible flow field whose quality is comparable to the iterative methods (Mansi et al., 2011; Xing et al., 2017; Yu et al., 2023), while being much faster"
- Please explain: "We average over the three magnitude images to obtain $I_{Mag}$"
- the use of (monomodal) multi-channel registration is not novel and warrants less emphasis in the paper
- "Determinant-based approaches focus on constraining the deformation field with a penalty on the Jacobian determinant of the deformation" deviating from unity.

Possible methodological improvements
- The proposed incompressibility loss term is highly asymmetric, penalizing shrinkage more than expansion. This might be beneficial to obtain diffeomorphic warps as it discourages sign-flips. However, it is likely leading to a bias towards preferentially tissue expansion - at least during initial training. Do you see such a bias in the final data?
  I suggest symmetrizing the loss, for instance with $|\log(\max(\det(J), \epsilon))| +  |\log(\max(2 - \det(J), \epsilon))|$ ([plot](https://www.wolframalpha.com/input?i=+abs%28log%28max%28x%2C+1e-3%29%29%29+%2B++abs%28log%28max%282+-+x%2C+1e-3%29%29%29++%2C+x+in%5B-1%2C+3%5D)) or using a general purpose robust loss function of the deviation of the Jacobian determinant from unity instead.
- Is a diffeomorphic algorithm, encouraged by scaling and squaring layer, and unit Jacobian determinant and smoothness regulation a good choice for the tongue that glides over adjacent tissue or would a segmentation step that isolates the tongue similar to DOI:10.1016/j.compmedimag.2014.07.004 be beneficial? You could use the motion sequence to learn this and then apply the regularization only to the tongue.
- I would suggest weighting the smoothness constraint also by $I_{Mag}$ to allow large shear in air-filled regions when the tongue glides.
- How does performance on the patient population change if you also train on the pathological subjects?

**Paper Type:**

methodological development

**Questions To Address In The Rebuttal:**

- Add context of prior work and related imaging techniques for motion estimation.
- Assess performance against other algorithms and on other organs or on simulated data.
- Improve the evaluation of incompressibility.
- The method claims novelty for introducing a custom weighting of the Jacobian determinant to achieve incompressibility. Please comment on the properties of the proposed loss term and check whether it outperforms standard robust loss functions such as m-estimators or [arXiv:1701.03077](http://arxiv.org/abs/1701.03077).

---

### Official Review · Reviewer_vvNu · 2023-02-04

**Confidence:** 3
**Preliminary Rating:** 4
**Recommendation:** Poster

**Summary:**

In this paper, a deep-learning-based image registration framework is presented for motion estimation for tongue movement during speaking with tagged MRI. An incompressible constraint is introduced to preserve the volume of the tongue. The experiments (on a small dataset) indicate good performance of the presented method.

**Strengths:**

- The authors give a very detailed introduction to the problem, the clinical need, the background, and the related work.
- An interesting topic of motion estimation for tongue movement during speaking with tagged MRI is addressed.
- The paper presents an unsupervised deep-learning-based framework that is capable of accurately registering the images.
- an incompressible constraint is introduced and used to preserve the volume of the tongue and therefore generate a more realistic deformation field


**Weaknesses:**

- The discussion including the limitations of the presented method is completely missing.
- It is not clear if the data split is performed subject-wise or subject-phrase-wise.
- In the related work section, the authors are partially too harsh with the limitations of other work. Please present them in a fair way. Your paper still adds value; therefore, you don’t need to do this.
- The experiments are performed on a very limited dataset.


**Deanonymize Review:**

no

**Detailed Comments:**

- “ However, these methods require the acquisition of a large number of closely spaced image slices, making them impractical for routine clinical use due to a large amount of time required.” How long does the acquisition required methods take compared to others?

- “require manual tuning for each new image pair.” This is not true. It is correct that a tuning step of the hyperparameters is required for a specific task. Typically, that should be enough, and the parameters don’t have to be tuned for every image pair. Of course, you could further improve the results if you do this.
- “However, these approaches are limited by their reliance on only two scalar images—" I would argue that this statement is wrong. The cited papers are just using two images, however, the approaches can easily be extended to more than two images.

- “ these approaches are unable to preserve the incompressibility of the motion field” Also for this statement I would argue that the approach is able but didn’t intend to preserve the incompressibility of the motion field.

- In Hering, A., Häger, S., Moltz, J., Lessmann, N., Heldmann, S., & van Ginneken, B. (2021). CNN-based lung CT registration with multiple anatomical constraints. Medical Image Analysis, 72, 102139. , the authors introduced a similar volume change constraints as used in this paper here.


- The Learn2Reg image challenge paper gives a good overview of current image registration approaches and might be a good reference.

- “The HC data was split into training, validation, and test datasets in a ratio of 0.6:0.2:0.2; patient data was reserved for testing.” –> “the post-partial glossectomy patient data” would be clearer.

- “ A scaling and squaring layer (Dalca et al., 2018) is used as the final layer to encourage our models to be diffeomorphic.” What is the effect of this layer in comparison to the introduced volume-preserving loss term? In table 1, the authors show that there is still neg. Jacobian Determinant when only using the scaling and squaring layer, but that means that the number of iterations is just not high enough. Is that correct? How long does it take to perform as many iterations as needed to not have any foldings left? How does this affect the average of the Jacobian Determinant?

- “It is often difficult to obtain the true dense motion field for evaluating the accuracy of registration algorithms. However, we can use the harmonic phase as a ground truth, as it is a property of tissue that moves with the tissue.” This explanation is not clear to me. The harmonic phase is not used as “the true dense motion field for evaluating the accuracy of registration algorithms” – correct? Please rephrase those sentences to make them easier to understand.

- Have you performed the Wilcoxon rank sum test on the 2 pathological cases? I cannot believe that you can reach significance with just 2 cases. If you use multiple subject-phrase as samples in the test, the samples aren’t independent.


**Paper Type:**

both

**Questions To Address In The Rebuttal:**

Please add a fair discussion including limitations of the proposed work and address the points listed in detailed comments to further improve your paper.

You could also add a few result images into the appendix and the best, average, and worse case. That helps the reader to better understand the outcome/results

---

### Meta-Review · Area_Chair_HTnQ · 2023-02-23

**Recommendation:** Accept (Oral)
**Confidence:** 5

**Metareview:**

All reviewers recommended acceptance of this manuscript. Overall, the problem of training an unsupervised registration model on motion estimation for tongue movement during speaking with limited data of tagged MRIs is challenging and interesting. The proposed method is fairly new and will be of a good interest to MIDL community. It will bring nice discussions in the conference. However, the authors shall carefully address all questions and concerns (as discussed in the rebuttal) in a revised version for final publication.